# AdaptSSR: Pre-training User Model with Augmentation-Adaptive Self-Supervised Ranking

**Yang Yu**[1,2], **Qi Liu**[1,2*], **Kai Zhang**[1,2], **Yuren Zhang**[1,2], **Chao Song**[3], **Min Hou**[4]
**Yuqing Yuan**[3], **Zhihao Ye**[3], **Zaixi Zhang**[1,2], **Sanshi Lei Yu**[1,2]
[1]Anhui Province Key Laboratory of Big Data Analysis and Application,
University of Science and Technology of China
[2]State Key Laboratory of Cognitive Intelligence
[3]OPPO Research Institute  [4]Hefei University of Technology
{yflyl613, kkzhang0808, yr160698, zaixi}@mail.ustc.edu.cn
{songchao12, yuanyuqing, yezhihao3}@oppo.com
qiliuql@ustc.edu.cn, {hmhoumin, meet.leiyu}@gmail.com

## Abstract

User modeling, which aims to capture users' characteristics or interests, heavily relies on task-specific labeled data and suffers from the data sparsity issue. Several recent studies tackled this problem by pre-training the user model on massive user behavior sequences with a contrastive learning task. Generally, these methods assume different views of the same behavior sequence constructed via data augmentation are *semantically consistent*, i.e., reflecting similar characteristics or interests of the user, and thus maximizing their agreement in the feature space. However, due to the diverse interests and heavy noise in user behaviors, existing augmentation methods tend to lose certain characteristics of the user or introduce noisy behaviors. Thus, forcing the user model to directly maximize the similarity between the augmented views may result in a negative transfer. To this end, we propose to replace the contrastive learning task with a new pretext task: Augmentation-**Adapt**ive **S**elf-**S**upervised **R**anking (**AdaptSSR**), which alleviates the requirement of semantic consistency between the augmented views while pre-training a discriminative user model. Specifically, we adopt a multiple pairwise ranking loss which trains the user model to capture the similarity orders between the implicitly augmented view, the explicitly augmented view, and views from other users. We further employ an in-batch hard negative sampling strategy to facilitate model training. Moreover, considering the distinct impacts of data augmentation on different behavior sequences, we design an augmentation-adaptive fusion mechanism to automatically adjust the similarity order constraint applied to each sample based on the estimated similarity between the augmented views. Extensive experiments on both public and industrial datasets with six downstream tasks verify the effectiveness of AdaptSSR.

## 1 Introduction

User modeling aims to capture the user's characteristics or interests and encode them into a dense user representation for a specific user-oriented task, such as user profiling, personalized recommendation, and click-through rate prediction [24, 39, 50, 59]. In the past decade, extensive methods [5, 9, 12, 22, 53] have been proposed to model users based on their historical behaviors. Despite their great success in various tasks, these methods usually rely on a large amount of task-specific labeled data to train accurate user models, which makes them suffer from the data sparsity problem [17, 36, 57].

---

*Qi Liu is the corresponding author.

37th Conference on Neural Information Processing Systems (NeurIPS 2023).

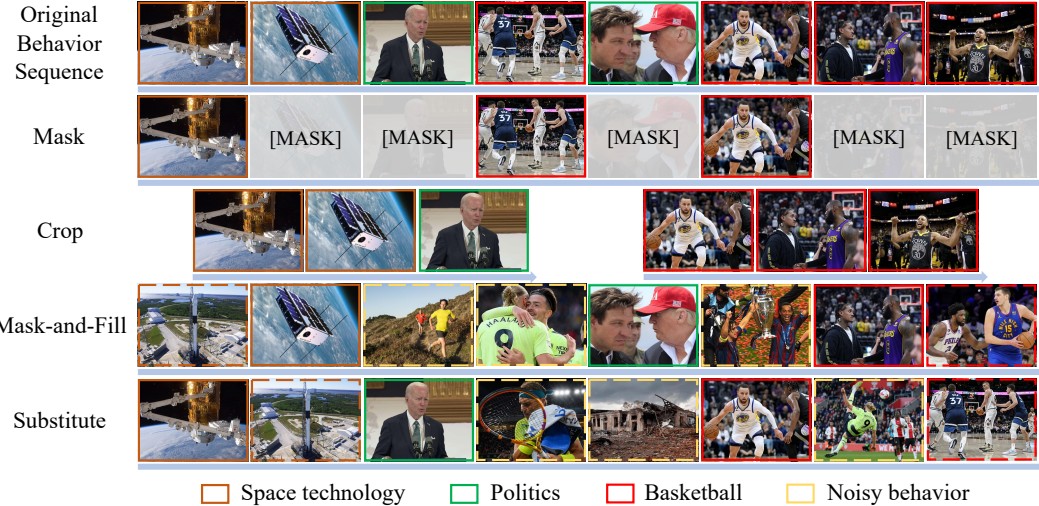

Figure 1: An illustration of the impact of different data augmentation methods on the user behavior sequence. For a more intuitive demonstration, we use a picture to represent the main topic of each news article clicked by the user. Pictures with the same color border reflect similar interests of the user. Dash borders are used to indicate behaviors replaced by the data augmentation method.

A mainstream technique to tackle this challenge is the pre-training paradigm [33, 46, 54, 56]. The user model is first pre-trained on a mass of unlabeled user behavior data, which enables it to extract users' general characteristics from their historical behaviors. Then the model is transferred to benefit various downstream tasks via fine-tuning. Motivated by the recent development in CV [3, 7, 14] and NLP [13, 23], several works [2, 8, 38] explore pre-training the user model with a contrastive learning task. They construct different views of the same behavior sequence via data augmentation and assume them to be *semantically consistent*, i.e., reflecting similar characteristics or interests of the user [27, 47]. Thus, these views are regarded as positive samples and a contrastive loss (e.g., InfoNCE [30]) is further applied to maximize their similarity in the feature space.

Although such pre-trained user models have proven to benefit various downstream tasks, we find that their assumption about semantic consistency between the augmented views does not necessarily hold. Unlike texts and images, user behaviors usually contain not only diverse interests but heavy noise as well [4, 26, 43]. Hence, existing data augmentation methods tend to result in augmented sequences that lose certain characteristics of the user or introduce noisy interests that the user does not have. To illustrate the impact of different augmentation methods on the behavior sequence, we randomly sample a user from the MSN online news platform[2]. His recent eight behaviors and the results of several data augmentation methods are shown in Fig. 1. From the original behavior sequence, we may infer the user's interests in space technology, politics, and basketball. Meanwhile, we find that various augmentation methods have distinct impacts on different behavior sequences. For example, the masked behavior sequence [47] loses the user's interest in politics due to randomness, and the two sequences augmented by cropping [38] reflect the user's different interests over time. Other methods such as mask-and-fill [2] and substitute [27] bring in behaviors related to football and tennis which the user may not be actually interested in. However, existing contrastive learning-based methods force the user model to maximize the agreement between such augmented views no matter whether they are similar or not, which may lead to a negative transfer for the downstream task.

To address this problem, in this paper, instead of trying to design a new data augmentation method that guarantees to generate consistent augmented behavior sequences, we escape the existing contrastive learning framework and propose a new pretext task: Augmentation-**Adapt**ive **S**elf-**S**upervised **R**anking (**AdaptSSR**), which alleviates the requirement of semantic consistency between the augmented views while pre-training a discriminative user model. Since the augmented views may reflect distinct characteristics or interests of the user, instead of directly maximizing their agreement, we adopt a multiple pairwise ranking loss which trains the user model to capture the similarity orders between the implicitly augmented view, the explicitly augmented view, and views from other users.

---

[2]More examples are provided in Appendix A.

Specifically, the user model is trained to project the original view closer to the implicitly augmented view than views from other users in the feature space, with the explicitly augmented view in between. We further utilize an in-batch hard negative sampling strategy to facilitate model training. In addition, since data augmentation has distinct impacts on different behavior sequences, we design an augmentation-adaptive fusion mechanism to automatically adjust the similarity order constraint applied to each sample based on the semantic similarity between the augmented views. Extensive experiments on both public and industrial datasets with six downstream tasks validate that AdaptSSR consistently outperforms existing pre-training methods while adapting to various data augmentation methods. Our code is available at `https://github.com/yflyl613/AdaptSSR`.

## 2 Related Work

**Contrastive Learning** Aiming to learn high-quality representations in a self-supervised manner, contrastive learning has achieved great success in various domains [1, 6, 28, 41]. Its main idea is to learn a mapping function that brings positive samples closer together while pushing negative samples further apart in the feature space. One of the key issues for contrastive learning is how to construct semantically consistent positive samples. To tackle this problem, several data augmentation methods such as jittering, rotation, and multi-cropping have been proposed for images [3, 16, 29]. However, for user behavior sequences, due to the diverse user interests and heavy noise, it is hard to determine their semantics and even harder to provide a semantically consistent augmentation [25, 32, 49], which limits the effectiveness of existing contrastive learning-based user model pre-training methods.

**User Model Pre-training** Various methods [24, 53, 60] have been proposed to model users based on their behaviors and task-specific labeled data. However, these supervised methods usually encounter data sparsity issues [18, 36]. Inspired by the success of the pre-training paradigm in CV and NLP, several studies [15, 33, 46] have sought to empower various user-oriented tasks by pre-training the user model on unlabeled behavior sequences. Predominant methods can be categorized into two groups: generative and discriminative. Generative methods [45, 52] usually first corrupt the behavior sequence and then try to restore the masked part. However, they largely focus on mining the correlation between behaviors but lack holistic user representation learning, which limits their effectiveness in user modeling [8]. Recently, several discriminative methods [2, 38] have applied contrastive learning to pre-train the user model. They assume that different views of the same behavior sequence constructed via data augmentation are semantically similar and take them as the positive sample. In contrast, we propose a self-supervised ranking task that avoids directly maximizing the similarity between the augmented views and alleviates the requirement of their semantic consistency.

**Data Augmentation for User Behavior Sequences** In the domain of user modeling and sequential recommendation, numerous data augmentation methods [2, 8, 38, 47] have been proposed for user behavior sequences, which can be broadly categorized into two groups: explicit and implicit. Explicit augmentation methods, such as masking and cropping [27, 47], are performed at the data level, meaning that they directly modify the behavior sequence. On the other hand, implicit augmentation is performed at the model level. The same sequence is input into the model twice with different independently sampled dropout masks, which adds distinct noise in the feature space [8, 32]. Our method utilizes both implicitly and explicitly augmented views and trains the user model to capture their similarity order. Moreover, owing to the augmentation-adaptive fusion mechanism, our AdaptSSR can adapt to diverse data augmentation methods and consistently improve their effectiveness.

## 3 Methodology

In this section, we introduce the details of our Augmentation-Adaptive Self-Supervised Ranking (AdaptSSR) task for user model pre-training. Fig. 2 illustrates the framework of AdaptSSR and the pseudo-codes for the entire pre-training procedure are provided in Appendix B.

### 3.1 Notations and Problem Statement

Suppose that we have collected a large number of user behavior data from a source domain, e.g., an online news platform or an e-commerce platform. Let $\mathcal{U}$ and $\mathcal{I}$ denote the set of users and different behaviors, where each behavior $x \in \mathcal{I}$ can be a news article or an item. It should be noted

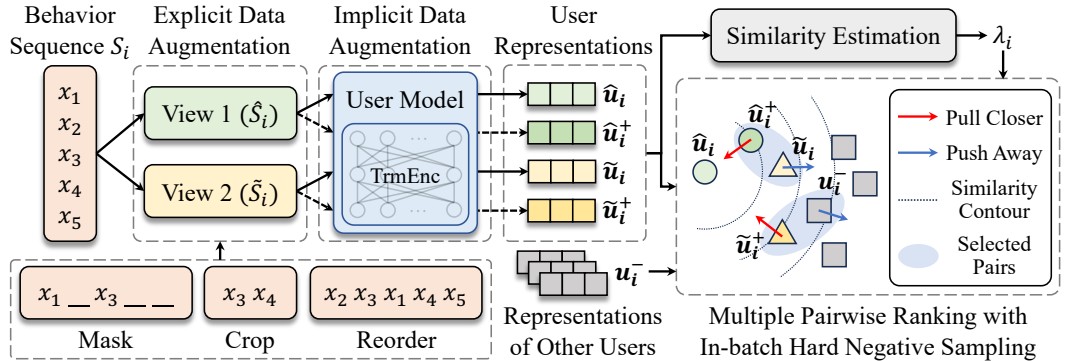

Figure 2: The framework of AdaptSSR. A sequence with five user behaviors is used for illustration.

that our methodology aligns with the prevalent settings in many previous works [2, 8, 47, 52] and real-world scenarios, where each behavior is simply represented by an ID. We refrain from using any ancillary information related to user behaviors (such as images or text descriptions). For each user $u \in \mathcal{U}$, we organize his/her behaviors in chronological order and denote the behavior sequence as $S = \{x_1, x_2, \ldots, x_n\}$ ($x_i \in \mathcal{I}$). Our goal is to pre-train a user model based on these behavior sequences. The output of the user model is a dense user representation $u \in \mathbb{R}^d$ reflecting the general characteristics of the user, where $d$ is the embedding dimension. For each downstream task, a simple MLP is added and the entire model is fine-tuned with the labeled data of the target task.

## 3.2 Self-Supervised Ranking

As it is hard to perform a semantically consistent augmentation for the user behavior sequence, we avoid directly maximizing the agreement between the augmented views and propose a self-supervised ranking task that trains the user model to capture the similarity order between the implicitly augmented view, the explicitly augmented view, and views from others. Specifically, given a behavior sequence $S$, we first apply the implicit data augmentation, i.e., the same sequence is input into the user model twice with different independently sampled dropout masks, and denote the generated user representations as $u$ and $u^+$. Then, we apply certain explicit data augmentation to $S$ and denote the augmented sequence and its corresponding user representation as $\hat{S}$ and $\hat{u}$, respectively. Since $u$ differs from $u^+$ only because of the distinct dropout masks applied by the implicit augmentation, while the difference between $u$ and $\hat{u}$ is caused by both the implicit augmentation and the explicit augmentation which modifies $S$ at the data level, we require them to satisfy the following similarity order: $\text{sim}(u, u^+) \geqslant \text{sim}(u, \hat{u})$. Considering that the semantic consistency between $S$ and $\hat{S}$ cannot be guaranteed, our pre-training objective is to capture the following similarity order:

$$\boldsymbol{\Gamma} : \text{sim}(u, u^+) \geqslant \text{sim}(u, \hat{u}) \geqslant \text{sim}(u, u^-), \tag{1}$$

where $u^-$ is the user representation of another user. Instead of directly forcing the user model to maximize $\text{sim}(u, \hat{u})$ as existing contrastive learning-based methods, our objective leaves room for the user model to adjust $\text{sim}(u, \hat{u})$ properly between the upper bound $\text{sim}(u, u^+)$ and the lower bound $\text{sim}(u, u^-)$ based on the similarity between the augmented views, which alleviates the requirement of their semantic consistency.

## 3.3 Implementation

**User Model**   Our method is decoupled from the structure of the user model, allowing for the integration of various existing methods, such as the attention network [58] and the temporal convolutional network [51]. In this work, we choose the widely used Transformer Encoder (TrmEnc) [40] as the backbone of the user model due to its superior performance in sequence modeling [20, 37]. Given a user behavior sequence $S = \{x_1, x_2, \ldots, x_n\}$, each behavior $x_i \in \mathcal{I}$ is projected into a dense vector $x_i \in \mathbb{R}^d$ via looking-up an embedding table $\mathbf{E} \in \mathbb{R}^{|\mathcal{I}| \times d}$. Another learnable position embedding matrix $\mathbf{P} = [p_1, p_2, \ldots, p_n] \in \mathbb{R}^{n \times d}$ is used to model the position information. An attention network with a learnable query vector $q \in \mathbb{R}^d$ is further utilized to aggregate the output

hidden states $\mathbf{H}$ of the Transformer Encoder. Thus, the user representation is calculated as follows:

$$\mathbf{H} = [\boldsymbol{h}_1, \boldsymbol{h}_2 \ldots, \boldsymbol{h}_n] = \text{TrmEnc}([\boldsymbol{x}_1 + \boldsymbol{p}_1, \boldsymbol{x}_2 + \boldsymbol{p}_2, \ldots, \boldsymbol{x}_n + \boldsymbol{p}_n]), \tag{2}$$

$$\alpha_i = \frac{\exp(\boldsymbol{q}^\top \boldsymbol{h}_i)}{\sum_{j=1}^{n} \exp(\boldsymbol{q}^\top \boldsymbol{h}_j)}, \tag{3}$$

$$\boldsymbol{u} = \sum_{i=1}^{n} \alpha_i \boldsymbol{h}_i. \tag{4}$$

**Multiple Pairwise Ranking with In-batch Hard Negative Sampling**  Given a batch of user behavior sequences $\{S_i\}_{i=1}^{B}$, where $B$ is the batch size, we first apply two randomly selected explicit augmentation methods to each $S_i$ and denote the augmented sequence as $\hat{S}_i$ and $\tilde{S}_i$. In this work, we adopt the three random augmentation operators proposed by Xie et al. [47]: mask, crop, and reorder. Notably, our method can be integrated with various existing data augmentation methods, which will be further explored in Section 4.3. Then, for a pair of $\hat{S}_i$ and $\tilde{S}_i$, we input each of them into the user model twice with independently sampled dropout mask and denote the generated user representations as $\hat{\boldsymbol{u}}_i, \hat{\boldsymbol{u}}_i^+$ and $\tilde{\boldsymbol{u}}_i, \tilde{\boldsymbol{u}}_i^+$, respectively. It should be noted that we only utilize the standard dropout mask in the Transformer Encoder, which is applied to both the attention probabilities and the output of each sub-layer. We treat all representations generated from views of other users in the same batch as negative samples, which are denoted as $\mathbf{U}_i^- = \{\hat{\boldsymbol{u}}_j, \hat{\boldsymbol{u}}_j^+, \tilde{\boldsymbol{u}}_j, \tilde{\boldsymbol{u}}_j^+\}_{j=1, j \neq i}^{B}$. Inspired by previous works in learning to rank, we adopt a Multiple Pairwise Ranking (MPR) loss [48] to train the user model to capture the similarity order $\boldsymbol{\Gamma}$, which extends the BPR loss [34] to learn two pairwise ranking orders simultaneously. For the user representation $\hat{\boldsymbol{u}}_i$ and $\hat{\boldsymbol{u}}_i^+$, each $\boldsymbol{v} \in \{\tilde{\boldsymbol{u}}_i, \tilde{\boldsymbol{u}}_i^+\}$ and $\boldsymbol{w} \in \mathbf{U}_i^-$ form a quadruple for model training, and the loss function for $\hat{S}_i$ is formulated as follows:

$$\hat{\mathcal{L}}_i = -\frac{1}{2|\mathbf{U}_i^-|} \sum_{\boldsymbol{v} \in \{\tilde{\boldsymbol{u}}_i, \tilde{\boldsymbol{u}}_i^+\}} \sum_{\boldsymbol{w} \in \mathbf{U}_i^-} \log \sigma \big[ \lambda \left( \text{sim}(\hat{\boldsymbol{u}}_i, \hat{\boldsymbol{u}}_i^+) - \text{sim}(\hat{\boldsymbol{u}}_i, \boldsymbol{v}) \right)$$
$$+ (1 - \lambda) \left( \text{sim}(\hat{\boldsymbol{u}}_i, \boldsymbol{v}) - \text{sim}(\hat{\boldsymbol{u}}_i, \boldsymbol{w}) \right) \big], \tag{5}$$

where $\sigma(\cdot)$ is the Sigmoid function and $\text{sim}(\cdot)$ calculates the cosine similarity between two vectors. $\lambda \in [0, 1]$ is a trade-off hyper-parameter for fusing the two pairwise ranking orders: $\text{sim}(\hat{\boldsymbol{u}}_i, \hat{\boldsymbol{u}}_i^+) \geqslant \text{sim}(\hat{\boldsymbol{u}}_i, \boldsymbol{v})$ and $\text{sim}(\hat{\boldsymbol{u}}_i, \boldsymbol{v}) \geqslant \text{sim}(\hat{\boldsymbol{u}}_i, \boldsymbol{w})$.

In Eq. (5), each quadruple is treated equally. Inspired by recent studies [11, 44, 55] on the impact of hard negative samples for contrastive learning, we further design an in-batch hard negative sampling strategy to facilitate model training. For each pairwise ranking order, we select the pair with the smallest similarity difference for training, i.e., the loss function is formulated as follows:

$$\hat{\mathcal{L}}_i = -\log \sigma \left[ \lambda \left( \text{sim}(\hat{\boldsymbol{u}}_i, \hat{\boldsymbol{u}}_i^+) - \max_{\boldsymbol{v} \in \{\tilde{\boldsymbol{u}}_i, \tilde{\boldsymbol{u}}_i^+\}} \text{sim}(\hat{\boldsymbol{u}}_i, \boldsymbol{v}) \right) \right.$$
$$\left. + (1 - \lambda) \left( \min_{\boldsymbol{v} \in \{\tilde{\boldsymbol{u}}_i, \tilde{\boldsymbol{u}}_i^+\}} \text{sim}(\hat{\boldsymbol{u}}_i, \boldsymbol{v}) - \max_{\boldsymbol{w} \in \mathbf{U}_i^-} \text{sim}(\hat{\boldsymbol{u}}_i, \boldsymbol{w}) \right) \right]. \tag{6}$$

Such pairs are currently the closest samples that the model needs to discriminate, which can provide the most significant gradient information during training. For another augmented sequence $\tilde{S}_i$, the loss function $\tilde{\mathcal{L}}_i$ is symmetrically defined and the overall loss is computed as $\mathcal{L} = \sum_{i=1}^{B} (\hat{\mathcal{L}}_i + \tilde{\mathcal{L}}_i)/2B$.

**Augmentation-Adaptive Fusion**  With the loss function $\hat{\mathcal{L}}_i$, the user model is trained to capture two pairwise ranking orders: $\text{sim}(\hat{\boldsymbol{u}}_i, \hat{\boldsymbol{u}}_i^+) \geqslant \text{sim}(\hat{\boldsymbol{u}}_i, \boldsymbol{v})$ and $\text{sim}(\hat{\boldsymbol{u}}_i, \boldsymbol{v}) \geqslant \text{sim}(\hat{\boldsymbol{u}}_i, \boldsymbol{w})$ simultaneously. However, we further need to combine them properly via the hyper-parameter $\lambda$. As illustrated in Fig. 1 and Appendix A, the effects of data augmentation vary significantly across different behavior sequences. Consequently, a fixed and unified $\lambda$ is not enough to fuse the two pairwise ranking orders properly for different samples. Thus, we further develop an augmentation-adaptive fusion mechanism to automatically adjust the similarity order constraint applied to each sample based on the similarity between the augmented views. Specifically, we estimate the semantic similarity between the explicitly augmented views with the average similarity between the user representations generated from $\hat{S}_i$ and

$\tilde{S}_i$. Then we replace the hyper-parameter $\lambda$ in Equation (6) with a dynamic coefficient $\lambda_i$ for each training sample $S_i$, which is calculated along the training procedure as follows:

$$\lambda_i = 1 - \frac{1}{4} \sum_{\hat{\boldsymbol{s}} \in \{\hat{\boldsymbol{u}}_i, \hat{\boldsymbol{u}}_i^+\}} \sum_{\tilde{\boldsymbol{s}} \in \{\tilde{\boldsymbol{u}}_i, \tilde{\boldsymbol{u}}_i^+\}} \max(\mathrm{sim}(\hat{\boldsymbol{s}}, \tilde{\boldsymbol{s}}), 0). \tag{7}$$

To get an accurate similarity estimation at the beginning of the training, we train the user model with the MLM task [52] until convergence before applying our self-supervised ranking task. If $\hat{S}_i$ and $\tilde{S}_i$ are semantically similar, $\lambda_i$ will be small and set the loss function $\hat{\mathcal{L}}_i$ focusing on maximizing the latter term $\min_{\boldsymbol{v} \in \{\tilde{\boldsymbol{u}}_i, \tilde{\boldsymbol{u}}_i^+\}} \mathrm{sim}(\hat{\boldsymbol{u}}_i, \boldsymbol{v}) - \max_{\boldsymbol{w} \in \mathbf{U}_i^-} \mathrm{sim}(\hat{\boldsymbol{u}}_i, \boldsymbol{w})$, which forces the user model to discriminate these similar explicitly augmented views from views of other users. Otherwise, $\lambda_i$ will be large and train the user model to pull the implicitly augmented view and these dissimilar explicitly augmented views apart. As a result, the user model is trained to adaptively adjust $\mathrm{sim}(\hat{\boldsymbol{u}}_i, \boldsymbol{v})$ when combining the two learned pairwise ranking orders for each sample, which can better deal with the distinct impacts of data augmentation on different behavior sequences.

### 3.4 Discussion

In this subsection, we discuss the connection between our proposed self-supervised ranking task and existing contrastive learning-based methods.

If we set $\lambda_i \equiv 0$ for all training samples and input each behavior sequence into the user model once (i.e., do not apply the implicit data augmentation), our loss function $\hat{\mathcal{L}}_i$ degenerates as follows:

$$\begin{aligned}
\hat{\mathcal{L}}_i' &= -\log \sigma \left[ \mathrm{sim}(\hat{\boldsymbol{u}}_i, \tilde{\boldsymbol{u}}_j) - \max_{\boldsymbol{w} \in \mathbf{V}_i^-} \mathrm{sim}(\hat{\boldsymbol{u}}_i, \boldsymbol{w}) \right] \\
&= -\log \frac{\exp\left(\mathrm{sim}(\hat{\boldsymbol{u}}_i, \tilde{\boldsymbol{u}}_j)\right)}{\exp\left(\mathrm{sim}(\hat{\boldsymbol{u}}_i, \tilde{\boldsymbol{u}}_j)\right) + \max_{\boldsymbol{w} \in \mathbf{V}_i^-} \exp\left(\mathrm{sim}(\hat{\boldsymbol{u}}_i, \boldsymbol{w})\right)},
\end{aligned} \tag{8}$$

where $\mathbf{V}_i^- = \{\hat{\boldsymbol{u}}_j, \tilde{\boldsymbol{u}}_j\}_{j=1, j \neq i}^B$. Most existing contrastive learning-based pre-training methods [2, 32, 38] adopt the InfoNCE loss [30] to train the user model, which can be formulated as follows:

$$\mathcal{L}_{\mathrm{InfoNCE}} = -\log \frac{\exp\left(\mathrm{sim}\left(\hat{\boldsymbol{u}}_i, \tilde{\boldsymbol{u}}_j\right)\right)}{\exp\left(\mathrm{sim}\left(\hat{\boldsymbol{u}}_i, \tilde{\boldsymbol{u}}_j\right)\right) + \sum_{\boldsymbol{w} \in \mathbf{V}_i^-} \exp\left(\mathrm{sim}\left(\hat{\boldsymbol{u}}_i, \boldsymbol{w}\right)\right)}. \tag{9}$$

Both loss functions aim to maximize the agreement between the augmented views. The sole distinction is that $\hat{\mathcal{L}}_i'$ selects the hardest in-batch negative sample, which is most similar to the anchor $\hat{\boldsymbol{u}}_i$, whereas $\mathcal{L}_{\mathrm{InfoNCE}}$ uses the entire negative sample set $\mathbf{V}_i^-$. Thus, with the same data augmentation method, the degenerated version of AdaptSSR is equivalent to combining contrastive learning-based methods with hard negative sampling, which has been proven to be effective by several recent studies [19, 35, 42].

When $\lambda_i > 0$, the former term $\mathrm{sim}(\hat{\boldsymbol{u}}_i, \hat{\boldsymbol{u}}_i^+) - \max_{\boldsymbol{v} \in \{\tilde{\boldsymbol{u}}_i, \tilde{\boldsymbol{u}}_i^+\}} \mathrm{sim}(\hat{\boldsymbol{u}}_i, \boldsymbol{v})$ in $\hat{\mathcal{L}}_i$ forces the user model to capture the similarity order $\mathrm{sim}(\hat{\boldsymbol{u}}_i, \hat{\boldsymbol{u}}_i^+) \geqslant \mathrm{sim}(\hat{\boldsymbol{u}}_i, \boldsymbol{v})$ as well. In addition, our augmentation-adaptive fusion mechanism automatically adjusts the similarity order constraint applied to each sample. As a result, our method alleviates the requirement of semantic consistency between the augmented views and can adapt to various data augmentation methods.

## 4 Experiments

### 4.1 Experimental Setup

**Datasets**  We conduct experiments on two real-world datasets encompassing six downstream tasks. The first dataset, the Tencent Transfer Learning (TTL) dataset, was released by Yuan et al. [52] and contains users' recent 100 interactions on the QQ Browser platform. Additionally, it provides the downstream labeled data of two user profiling tasks: age prediction ($\mathcal{T}_1$) and life status prediction ($\mathcal{T}_2$), and two cold-recommendation tasks: click recommendation ($\mathcal{T}_3$) and thumb-up recommendation ($\mathcal{T}_4$). The second dataset, the App dataset, consists of users' app installation behaviors collected by a worldwide smartphone manufacturer, OPPO, from 2022-12 to 2023-03. Each user has been securely

Table 1: Detailed statistics of each dataset and downstream task.

| Dataset | TTL | | | | App | |
|---|---|---|---|---|---|---|
| # Behavior Sequences | 1,470,149 | | | | 1,575,837 | |
| # Different Behaviors | 645,972 | | | | 4,047 | |
| Avg. Sequence Length | 54.84 | | | | 44.13 | |
| Downstream Task | $\mathcal{T}_1$ | $\mathcal{T}_2$ | $\mathcal{T}_3$ | $\mathcal{T}_4$ | $\mathcal{T}_5$ | $\mathcal{T}_6$ |
| # Samples | 1,470,147 | 1,020,277 | 1,397,197 | 255,646 | 1,178,603 | 564,940 |
| # Labels/Items | 8 | 6 | 17,879 | 7,539 | 2 | 2 |

hashed into an anonymized ID to prevent privacy leakage. It also provides the downstream labeled data of a gender prediction task ($\mathcal{T}_5$), and a CVR prediction task ($\mathcal{T}_6$), which predicts whether the user will install a target app. Detailed statistics of each dataset and downstream task are listed in Table 1.

**Evaluation Protocols** For model pre-training, 90% user behavior sequences are randomly selected for training, while the rest 10% are used for validation. For each downstream task, we randomly split the dataset by 6:2:2 for training, validation, and testing. To evaluate model performance on various downstream tasks, we use classification accuracy (Acc) for multi-class classification tasks ($\mathcal{T}_1, \mathcal{T}_2$), NDCG@10 for cold-recommendation tasks ($\mathcal{T}_3, \mathcal{T}_4$), and AUC for binary classification tasks ($\mathcal{T}_5$, $\mathcal{T}_6$). To avoid the bias brought by sampled evaluation [21], we use the all-ranking protocol for the cold-recommendation task evaluation, i.e., all items not interacted by the user are used as candidates. We repeat each experiment 5 times and report the average results with standard deviations.

**Baselines** To validate the effectiveness of AdaptSSR, we compare it with three sets of baselines. The first set contains several generative pre-training methods, including:

- **PeterRec** [52] applies the MLM task [10] to the behavior sequence for user model pre-training.
- **PTUM** [45] pre-trains the user model with two self-supervised tasks: masked behavior prediction and next $K$ behavior prediction.

The second set contains several discriminative pre-training methods, including:

- **CLUE** [8] solely uses implicitly augmented views for contrastive pre-training.
- **CCL** [2] proposes a mask-and-fill strategy to construct high-quality augmented views with a data generator trained with the MLM task for contrastive pre-training.
- **IDICL** [38] takes different time periods of the same behavior sequence as augmented views for contrastive pre-training. An interest dictionary is used to extract multiple interests of the user.

The last set contains several data augmentation methods for user behavior sequences, including:

- **CL4SRec** [47] augments the behavior sequence by masking, cropping, and reordering.
- **CoSeRec** [27] takes item correlation into consideration and improves CL4SRec with two more informative augmentation operators: substitute and insert.
- **DuoRec** [32] takes sequences with the same last behavior as semantically similar positive samples.

We apply them to generate positive samples for the contrastive pre-training task. We also evaluate the performance of the model trained on the downstream task in an end-to-end manner. All these baselines share the same model structure with our AdaptSSR and only differ in the pre-training task.

**Implementation and Hyper-parameters** Following previous works [2, 8], we set the embedding dimension $d$ as 64. In the Transformer Encoder, the number of attention heads and layers are both set as 2. The dropout probability is set as 0.1. The data augmentation proportion $\rho$ for each baseline method is either searched from $\{0.1, 0.2, \ldots, 0.9\}$ or set as the default value in the original paper if provided. The batch size and learning rate are set as 128 and 2e-4 for both pre-training and fine-tuning. More implementation details are listed in Appendix C.

### 4.2 Overall Performance of Downstream Tasks

The experimental results on various downstream tasks are presented in Table 2. From the results, we have several findings. First, generative pre-training methods (PeterRec and PTUM) underperform compared to most discriminative pre-training methods. This is because these methods mainly focus

Table 2: Performance (%) of various pre-training methods on downstream tasks. Impr (%) indicates the relative improvement compared with the end-to-end training. The best results are **bolded**.

| Pre-train Method | $\mathcal{T}_1$ | | $\mathcal{T}_2$ | | $\mathcal{T}_3$ | | $\mathcal{T}_4$ | | $\mathcal{T}_5$ | | $\mathcal{T}_6$ | |
|---|---|---|---|---|---|---|---|---|---|---|---|---|
| | Acc | Impr | Acc | Impr | NDCG@10 | Impr | NDCG@10 | Impr | AUC | Impr | AUC | Impr |
| None | 62.87±0.05 | - | 52.24±0.16 | - | 1.99±0.03 | - | 2.87±0.07 | - | 78.63±0.06 | - | 75.14±0.14 | - |
| PeterRec | 63.62±0.11 | 1.19 | 53.14±0.07 | 1.72 | 2.37±0.02 | 19.10 | 3.06±0.08 | 6.62 | 79.61±0.13 | 1.25 | 76.04±0.10 | 1.20 |
| PTUM | 63.21±0.14 | 0.54 | 53.05±0.04 | 1.55 | 2.29±0.03 | 15.08 | 2.96±0.03 | 3.14 | 79.48±0.11 | 1.08 | 75.82±0.13 | 0.90 |
| CLUE | 63.38±0.10 | 0.81 | 53.23±0.05 | 1.90 | 2.38±0.02 | 19.60 | 3.05±0.21 | 6.27 | 79.90±0.06 | 1.62 | 76.03±0.16 | 1.18 |
| CCL | 63.76±0.11 | 1.42 | 53.37±0.09 | 2.16 | 2.43±0.02 | 22.11 | 3.32±0.13 | 15.68 | 80.22±0.07 | 2.02 | 77.35±0.10 | 2.94 |
| IDICL | 63.88±0.04 | 1.61 | 53.45±0.05 | 2.32 | 2.46±0.02 | 23.62 | 3.42±0.04 | 19.16 | 80.34±0.05 | 2.17 | 77.92±0.08 | 3.70 |
| CL4SRec | 63.71±0.14 | 1.34 | 53.43±0.05 | 2.28 | 2.41±0.03 | 21.11 | 3.29±0.06 | 14.63 | 80.14±0.08 | 1.92 | 77.02±0.05 | 2.50 |
| CoSeRec | 63.89±0.03 | 1.62 | 53.53±0.09 | 2.47 | 2.44±0.02 | 22.61 | 3.33±0.05 | 16.03 | 80.48±0.06 | 2.35 | 77.71±0.09 | 3.42 |
| DuoRec | 63.50±0.09 | 1.00 | 53.26±0.06 | 1.95 | 2.39±0.01 | 20.10 | 3.11±0.16 | 8.36 | 80.03±0.09 | 1.78 | 76.85±0.09 | 2.28 |
| **AdaptSSR** | **65.53±0.04** | **4.23** | **54.41±0.02** | **4.15** | **2.61±0.03** | **31.16** | **3.73±0.03** | **29.97** | **82.30±0.03** | **4.67** | **79.92±0.05** | **6.36** |

on mining the correlation between behaviors while lacking careful design for user representation learning, which limits their performance on downstream tasks. Second, discriminative pre-training methods with explicit data augmentation (e.g., CCL, CL4SRec, CoSeRec) generally outperform the method relying solely on implicit data augmentation (CLUE). We argue that this is because the implicit data augmentation caused by the dropout mask alone is too weak. The user model can easily distinguish the positive sample from others, thus providing limited knowledge for downstream tasks. Third, our AdaptSSR consistently surpasses previous SOTA contrastive learning-based pre-training methods by 2.6%, 1.7%, 6.1%, 9.1%, 2.3%, and 2.6% on each downstream task respectively, and our further t-test results show the improvements are significant at $p < 0.01$. This is because we train the user model to capture the similarity order $\Gamma$, rather than directly maximizing the similarity between the explicitly augmented views. Such a ranking task alleviates the requirement of semantic consistency while maintaining the discriminability of the pre-trained user model.

### 4.3 Performance with Different Data Augmentation Methods

As our method alleviates the requirement of semantic consistency between the augmented views and can adapt to a variety of data augmentation methods, we further combine it with several existing pre-training methods: CL4SRec, CoSeRec, and CCL, by replacing the contrastive learning (CL) task with our AdaptSSR while maintaining their data augmentation methods. We vary the augmentation proportion $\rho$ from 0.1 to 0.9 and evaluate the performance of these methods on the downstream age prediction task ($\mathcal{T}_1$). The results on other downstream tasks show similar trends and are included in Appendix D. From the results shown in Fig. 3, we find that these contrastive learning-based methods are quite sensitive to the data augmentation proportion. When $\rho$ is close to 0, the user model can easily discriminate these weakly augmented positive samples from others during pre-training and thus brings limited performance gain to the downstream task. However, the stronger the augmentation is, the more likely it is to generate dissimilar augmented views and may even cause a negative transfer to the downstream task. In contrast, our AdaptSSR significantly improves the performance of all these pre-training methods with

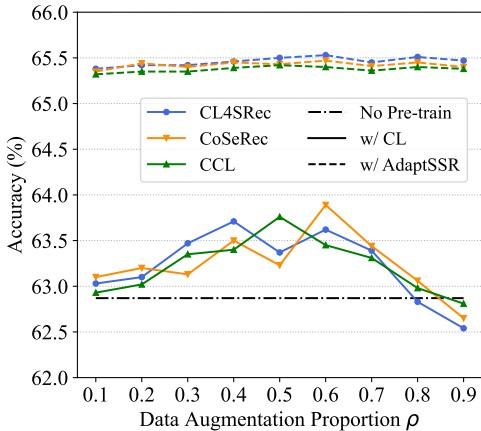

Figure 3: Effectiveness of AdaptSSR when combined with existing pre-training methods.

different augmentation proportions to a similar level. This is because our self-supervised ranking task takes the potential semantic inconsistency between the augmented views into account and avoids directly maximizing their similarity. In addition, our augmentation-adaptive fusion mechanism can properly combine the learned pairwise ranking orders based on the estimated similarity between the explicitly augmented views constructed by various augmentation methods with different strengths, which leads to similar model performance.

## 4.4 Ablation Study

In this subsection, we further validate the effectiveness of each component in our method, i.e., in-batch hard negative sampling, augmentation-adaptive fusion, and self-supervised ranking. We pre-train the user model with our AdaptSSR and its variant with one component removed. When the augmentation-adaptive fusion mechanism is removed, we search for the optimal $\lambda$ from $\{0, 0.1, ..., 1.0\}$. When the self-supervised ranking task is removed, we use the contrastive learning task instead. The results of the downstream age prediction task ($\mathcal{T}_1$) and thumb-up recommendation task ($\mathcal{T}_4$) are shown in Fig. 4. We first find that the model performance drops significantly by 2.8% on $\mathcal{T}_1$ and 11.8% on $\mathcal{T}_4$ when the self-supervised ranking task is removed. This is because it alleviates the requirement of semantic consistency between the augmented views, which is the major issue that impedes existing contrastive learning-based pre-training methods. Besides, the model performance also declines by 0.9%-1.6% on $\mathcal{T}_1$ and 3.1%-5.1% on $\mathcal{T}_4$ after we remove either in-batch hard negative sampling or augmentation-adaptive fusion. This is

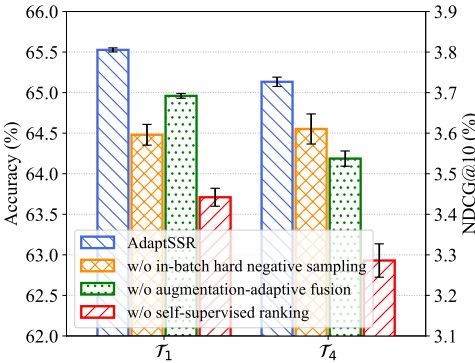

Figure 4: Effectiveness of each component in our AdaptSSR.

because in-batch hard negative sampling helps the model focus on discriminating the most similar samples while using all quadruples may introduce extra noise. Our augmentation-adaptive fusion mechanism adjusts the value of $\lambda_i$ based on the similarity between the augmented views for each training sample, while a constant $\lambda$ applies a fixed ranking order constraint to all samples and neglects the distinct impacts of data augmentation across different behavior sequences.

## 4.5 Efficiency Comparison

To compare the efficiency of different pre-training methods, we record their average training time on the TTL dataset and the App dataset and list them in Table 3. Our observations indicate that the average pre-training time of our AdaptSSR is similar to that of existing contrastive learning-based methods, such as CL4SRec and CoSeRec. Although our AdaptSSR requires inputting each behavior sequence into the model twice, we implement it by duplicating the input sequences in the batch size dimension. Thus, we only need to input all the sequences into the model once, which can be well parallelized by the GPU. Besides, for each pairwise ranking order, our in-batch hard negative sampling strategy only selects the pair with the smallest similarity difference to compute the multiple pairwise ranking loss, which avoids the costly softmax operation and cross-entropy loss calculation in the existing contrastive learning task. As a result,

Table 3: Average training time (hours) of different pre-training methods on each dataset.

| Pre-train Method | TTL | App |
|---|---|---|
| PeterRec | 2.927±0.022 | 1.537±0.010 |
| PTUM | 2.015±0.009 | 2.055±0.018 |
| CLUE | 1.453±0.016 | 1.633±0.015 |
| IDICL | 1.257±0.021 | 1.162±0.020 |
| CL4SRec | 1.868±0.013 | 2.081±0.017 |
| CoSeRec | 1.902±0.015 | 2.104±0.023 |
| DuoRec | 1.535±0.020 | 1.658±0.015 |
| AdaptSSR | 1.539±0.017 | 1.830±0.012 |

our AdaptSSR will not greatly increase the overall computational cost while bringing significant performance improvement to various downstream tasks.

## 4.6 User Representation Similarity Distribution Analysis

To better reflect the difference between our AdaptSSR and other pre-training methods, we randomly select 10,000 users from the TTL dataset. For each user's behavior sequence, we augment it by masking and cropping with a proportion of 0.6. Then, we calculate the cosine similarity between the user representations generated from the original behavior sequence, the augmented behavior sequence, and the behavior sequence of other users. We use kernel density estimation (KDE) [31] to estimate their probability density distributions, which are visualized in Fig. 5. The figure reveals that when pre-training with existing contrastive learning-based methods (e.g., CL4SRec, CoSeRec), the similarity between the augmented views always gathers in a very narrow range close to 1.0.

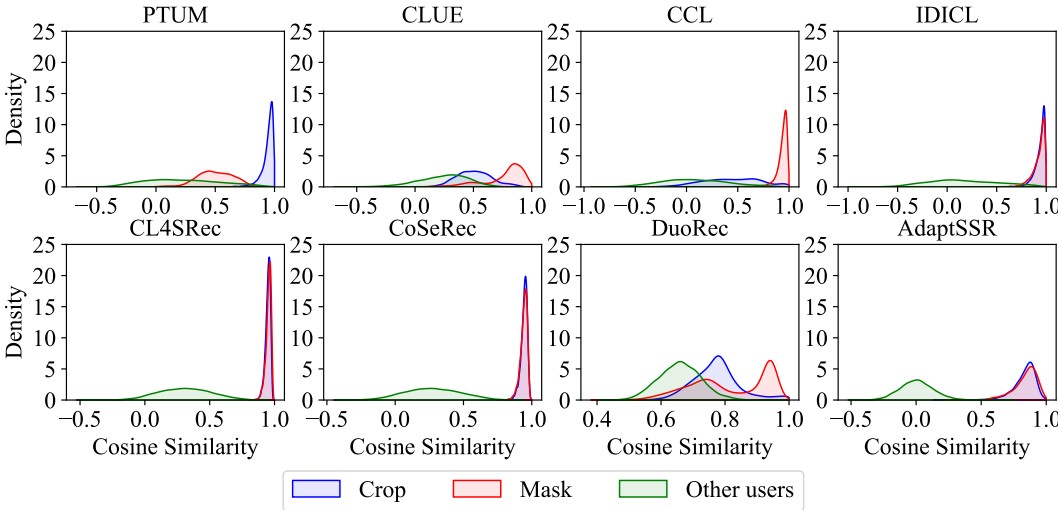

Figure 5: Distributions of the cosine similarity between user representations generated from the original behavior sequence, different augmented behavior sequences, and the behavior sequences of other users with various pre-training methods. The area under each curve equals to 1.

This is because these methods always maximize the similarity between the augmented views while neglecting their potential semantic inconsistency. Furthermore, when pre-trained with other methods (e.g., PTUM, CCL), the similarity between different users tends to disperse over a very broad range (from -0.5 to 1.0), which indicates that the user model cannot well distinguish the behavior sequence of different users. In contrast, the similarity between the augmented views distributes over a relatively wider range (from 0.5 to 1.0) when pre-trained with our AdaptSSR, which well fits our findings about these semantically inconsistent augmented views. Meanwhile, the distribution of the similarity between different users is still well separated from that between the augmented views, which means the user model can also discriminate the behavior sequence of different users.

## 5 Conclusion

In this paper, we identified the semantic inconsistency problem faced by existing contrastive learning-based user model pre-training methods. To tackle this challenge, we proposed a new pretext task: Augmentation-Adaptive Self-Supervised Ranking (AdaptSSR). A multiple pairwise ranking loss is adopted to train the user model to capture the similarity order between the implicitly augmented view, the explicitly augmented view, and views from other users. We further employed an in-batch hard negative sampling strategy to facilitate model training. Additionally, to cope with the distinct impacts of data augmentation on different behavior sequences, we designed an augmentation-adaptive fusion mechanism to adjust the similarity order constraint applied to each sample based on the estimated similarity between the augmented views. Our method can be further generalized to other domains where augmentation choices are not straightforward or could alter the semantics of the data. Extensive experiments validated that AdaptSSR consistently outperforms previous baselines by a large margin on various downstream tasks while adapting to diverse data augmentation methods.

## Acknowledgements

This research was partially supported by grants from the National Key Research and Development Program of China (No. 2021YFF0901003), and the OPPO joint research program. We furthermore thank the anonymous reviewers for their constructive comments.

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

# A    More Examples of the Semantic Inconsistency Problem

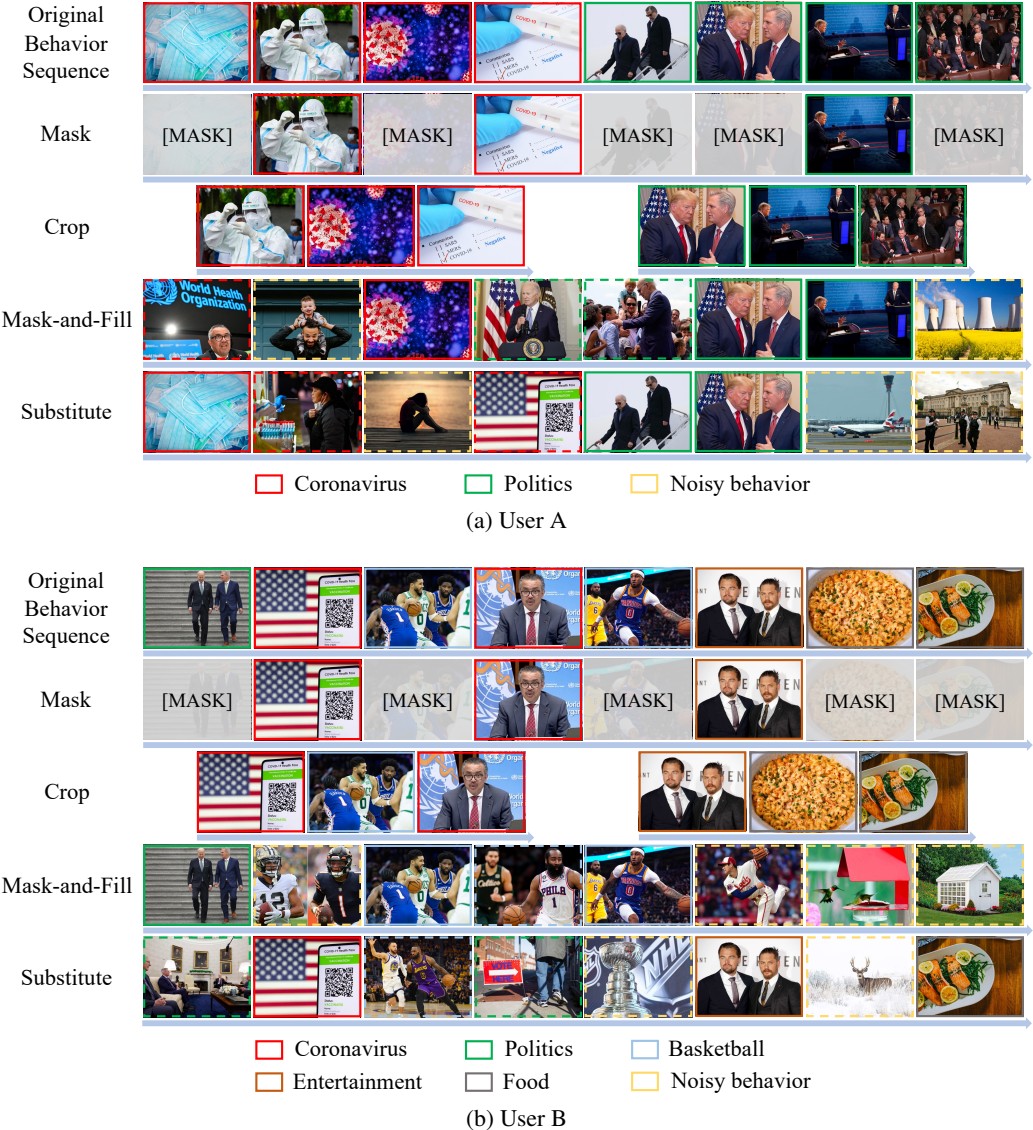

Figure 6: Illustrations of the impact of various data augmentation methods on different user behavior sequences. Each picture represents a news article clicked by the user. Pictures with the same color border reflect similar interests of the user. Dash borders are used to indicate behaviors replaced by the data augmentation method.

We randomly sampled two more anonymous users from the MSN online news platform to illustrate the impact of various data augmentation methods on different user behavior sequences and the consequent semantic inconsistency problem. The users' recent eight behaviors and the results of several data augmentation methods are shown in Fig. 6. The data augmentation proportion is set as 0.6. From the original behavior sequence of user A, we may infer that the user is quite concerned about coronavirus and politics. We also find that the behavior sequence augmented by masking well preserves the user's interests. However, the two sequences augmented by cropping contain completely different interests of the user over time. Meanwhile, some behaviors replaced by mask-and-fill and substitute are noisy, which the user may not be interested in. Similarly, the original behavior sequence of user B shows the user's potential interests in coronavirus, politics, basketball, entertainment, and food. As a result, augmentation methods such as mask and crop are likely to lose certain characteristics of the user, while augmentation methods such as mask-and-fill and substitute tend to bring in noisy behaviors. In

fact, it is hard to determine what are the real interests of user B from such a diverse behavior sequence since some behaviors may be the result of misclicking or click-baiting, thus making it even harder to provide a semantically consistent augmentation. From these examples, we can find that existing data augmentation methods may fail to preserve the characteristics or interests in the behavior sequence and cannot guarantee semantic consistency between the augmented views. Thus, directly forcing the user model to maximize the agreement between the augmented sequences may result in a negative transfer for downstream tasks.

## B The Algorithm of AdaptSSR

The pseudo-codes of the pre-training procedure with our AdaptSSR are shown in Algorithm 1.

---

**Algorithm 1** Pre-training Procedure with AdaptSSR

---

**Input:** A corpus of user behavior sequences $\mathcal{S}$ and a set of data augmentation operators $\mathcal{A}$.
**Output:** The pre-trained user model $\mathcal{M}$.
1: Train the user model $\mathcal{M}$ with the MLM task on $\mathcal{S}$ until convergence.
2: **while** not converged **do**
3:      Randomly sample a batch of user behavior sequences $\{S_i\}_{i=1}^B$ from $\mathcal{S}$.
4:      **for each** $S_i$ **do**
5:          Randomly select two augmentation operators $f$ and $g$ from $\mathcal{A}$.
6:          $\hat{\boldsymbol{u}}_i, \hat{\boldsymbol{u}}_i^+ \leftarrow \mathcal{M}(f(S_i)), \mathcal{M}(f(S_i))$.        ▷ With independently sampled dropout masks.
7:          $\tilde{\boldsymbol{u}}_i, \tilde{\boldsymbol{u}}_i^+ \leftarrow \mathcal{M}(g(S_i)), \mathcal{M}(g(S_i))$.        ▷ With independently sampled dropout masks.
8:          $\mathbf{U}_i^- \leftarrow \{\hat{\boldsymbol{u}}_j, \hat{\boldsymbol{u}}_j^+, \tilde{\boldsymbol{u}}_j, \tilde{\boldsymbol{u}}_j^+\}_{j=1, j\neq i}^B$.
9:          $\lambda_i \leftarrow 1 - \frac{1}{4} \sum_{\hat{\boldsymbol{s}} \in \{\hat{\boldsymbol{u}}_i, \hat{\boldsymbol{u}}_i^+\}} \sum_{\tilde{\boldsymbol{s}} \in \{\tilde{\boldsymbol{u}}_i, \tilde{\boldsymbol{u}}_i^+\}} \max(\text{sim}(\hat{\boldsymbol{s}}, \tilde{\boldsymbol{s}}), 0)$.
10:          $\hat{\mathcal{L}}_i \leftarrow -\log \sigma \Big[ \lambda_i \Big( \text{sim}(\hat{\boldsymbol{u}}_i, \hat{\boldsymbol{u}}_i^+) - \max_{\boldsymbol{v} \in \{\tilde{\boldsymbol{u}}_i, \tilde{\boldsymbol{u}}_i^+\}} \text{sim}(\hat{\boldsymbol{u}}_i, \boldsymbol{v}) \Big)$

               $+ (1 - \lambda_i) \Big( \min_{\boldsymbol{v} \in \{\tilde{\boldsymbol{u}}_i, \tilde{\boldsymbol{u}}_i^+\}} \text{sim}(\hat{\boldsymbol{u}}_i, \boldsymbol{v}) - \max_{\boldsymbol{w} \in \mathbf{U}_i^-} \text{sim}(\hat{\boldsymbol{u}}_i, \boldsymbol{w}) \Big) \Big]$.

11:          $\tilde{\mathcal{L}}_i \leftarrow -\log \sigma \Big[ \lambda_i \Big( \text{sim}(\tilde{\boldsymbol{u}}_i, \tilde{\boldsymbol{u}}_i^+) - \max_{\boldsymbol{v} \in \{\hat{\boldsymbol{u}}_i, \hat{\boldsymbol{u}}_i^+\}} \text{sim}(\tilde{\boldsymbol{u}}_i, \boldsymbol{v}) \Big)$

               $+ (1 - \lambda_i) \Big( \min_{\boldsymbol{v} \in \{\hat{\boldsymbol{u}}_i, \hat{\boldsymbol{u}}_i^+\}} \text{sim}(\tilde{\boldsymbol{u}}_i, \boldsymbol{v}) - \max_{\boldsymbol{w} \in \mathbf{U}_i^-} \text{sim}(\tilde{\boldsymbol{u}}_i, \boldsymbol{w}) \Big) \Big]$.

12:      **end for**
13:      $\mathcal{L} \leftarrow \sum_{i=1}^B (\hat{\mathcal{L}}_i + \tilde{\mathcal{L}}_i) / 2B$.
14:      Update the parameters of the user model $\mathcal{M}$ with $\mathcal{L}$ by backpropagation.
15: **end while**

---

## C Implementation Details

In our experiments, the embedding dimension $d$ is set as 64. In the Transformer Encoder, the number of attention heads and layers are both set as 2. We only utilize the standard dropout mask, which is applied to both the attention probabilities and the output of each sub-layer, and the dropout probability is set as 0.1. The maximum sequence length is set to 100 and 256 for the TTL dataset and the App dataset, respectively. For each downstream task, a two-layer MLP is added to the pre-trained user model, and the dimension of the intermediate layer is also set as 64.

For existing pre-training methods, the data augmentation proportion $\rho$ is either copied from previous works if provided or searched from $\{0.1, 0.2, \ldots, 0.9\}$. In our AdaptSSR, we adopt three random augmentation operators by default: mask, crop, and reorder. The augmentation proportion is set to 0.6, 0.4, and 0.6 respectively while our experimental results in Fig. 3 have shown that our method is robust to various data augmentation methods with different strengths.

We use the Adam optimizer for model training. The batch size and learning rate are set as 128 and 2e-4 for both pre-training and fine-tuning. The test results for all the models are reported at their best validation epoch. We repeat each experiment 5 times with different random seeds and report

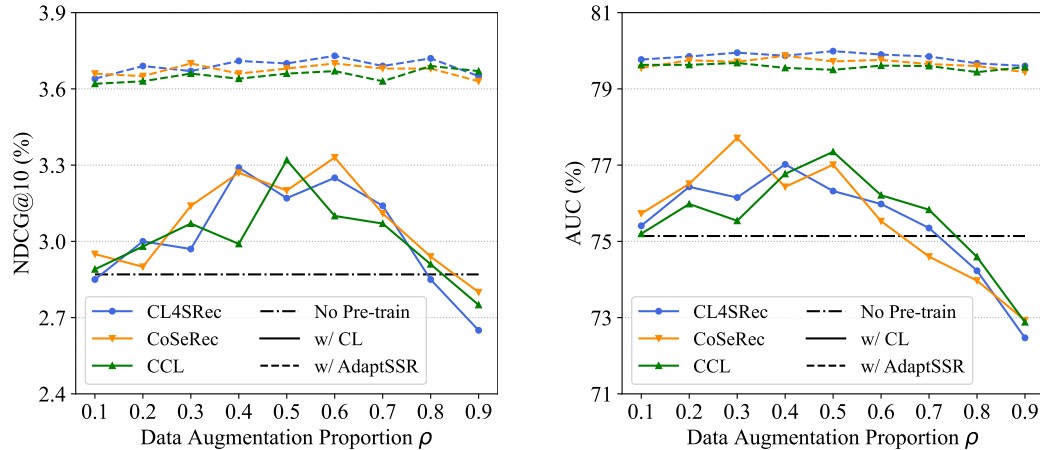

Figure 7: The effectiveness of AdaptSSR when combined with existing pre-training methods on the downstream thumb-up recommendation task (left) and CVR prediction task (right).

the average results with standard deviations. We implement all experiments with Python 3.8.13 and Pytorch 1.12.1 on an NVIDIA Tesla V100 GPU.

## D    More Experimental Results

Fig. 7 shows the performance of our AdaptSSR when combined with several existing pre-training methods: CL4SRec, CoSeRec, and CCL, on the downstream thumb-up recommendation task ($\mathcal{T}_4$) and CVR prediction task ($\mathcal{T}_6$) with the data augmentation proportion $\rho$ varying from 0.1 to 0.9. Similar to the results on the age prediction task ($\mathcal{T}_1$), we find that these contrastive learning-based methods are highly sensitive to the data augmentation proportion. A too-small or too-large value of $\rho$ will lead to limited performance gain or even negative transfer for the downstream task. Contrarily, our AdaptSSR consistently boosts the effectiveness of these pre-training methods by a large margin. This is because our self-supervised ranking task avoids directly maximizing the similarity between the augmented views. Moreover, our AdaptSSR also substantially improves the robustness of these pre-training methods to the augmentation proportion, which verifies the effectiveness of our augmentation-adaptive fusion mechanism. The dynamic coefficient $\lambda_i$ adjusts the similarity order constraint applied to each sample based on the similarity between the augmented views, thus empowering the pre-training methods to adapt to diverse data augmentation methods with varying strengths.

## E    Limitations

In this work, after pre-training the user model with our AdaptSSR, the entire model is fine-tuned with the downstream labeled data. However, fine-tuning such a large model for each downstream task can be time and space-consuming. A potential solution for this problem is Parameter-Efficient Fine-Tuning (PEFT). Several methods such as Adapter and LoRA have been demonstrated to be effective for adapting the large language model to various downstream tasks by only updating a small fraction of parameters. However, we empirically find that these methods perform poorly when applied to the pre-trained user model. A possible reason is that the currently widely used Transformer-based user model is much shallower and thinner than the large language model. Most of the model parameters belong to the bottom behavior embedding table, while the upper Transformer blocks only contain very few parameters. Since existing PEFT methods mainly focus on tuning parts of parameters in each Transformer block, only a very small fraction of parameters in total will be updated, which may not be enough to effectively adapt the user model to the downstream task. We will investigate how to transfer the pre-trained user model to various downstream tasks parameter-efficiently while maintaining the performance gain brought by AdaptSSR in our future work.

