# OpenReview forum: "AdaptSSR: Pre-training User Model with Augmentation-Adaptive Self-Supervised Ranking"
_NeurIPS.cc/2023/Conference — NeurIPS 2023 poster_

### Official Review · Reviewer_toy8 · 2023-06-28

**Soundness:** 3 good
**Presentation:** 3 good
**Contribution:** 2 fair
**Rating:** 5
**Confidence:** 4

**Summary:**

The paper tackles user-oriented tasks, e.g. personalized recommendation, and proposes a self-supervised method named AdaptSSR to replace the contrastive learning pre-training target task. It adopts a ranking loss that selects samples of smallest similarity differences and assigns dynamic weight coefficients to ranking parts based on the estimated similarity between the augmented views. Experiments on 6 downstream tasks from 2 datasets and several empirical analyses are conducted to verify the effectiveness of AdaptSSR.


**Strengths:**

The pre-training method and objective are clearly explained, and the equations in the text concisely demonstrate the proposed loss function.

**Weaknesses:**

The Multiple Pairwise Ranking loss, which is the core of the method, is not an original contribution of this paper, but an adaptation from Yu et al [49]. However, there is almost no mentioning of this work except for the source of the loss function, casting doubt on the novelty of the paper.


**Questions:**

Please elaborate on the differences between the approach proposed in this paper and [49], and highlight the contribution of this paper.

**Limitations:**

The authors have not discussed the limitations and broader societal impacts in the paper.

---

> ### Author Rebuttal · Authors · 2023-08-09
>
> We are sincerely grateful for the time and effort you have invested in reviewing our paper. In response to your insightful comments, we have provided detailed explanations and clarifications, which are enumerated below.
>
> **Weakness and Question 1**: The Multiple Pairwise Ranking loss, which is the core of the method, is not an original contribution of this paper, but an adaptation from Yu et al [49]. However, there is almost no mention of this work except for the source of the loss function, casting doubt on the novelty of the paper. Please elaborate on the differences between the approach proposed in this paper and [49], and highlight the contribution of this paper.
>
> **Response**: First, our work differs from Yu et al [49] in the following aspects:
>
> 1.  **The studied problems are completely different.** Yu et al [49] aim to train a better collaborative filtering model for item recommendation based on users' implicit feedback, while we aim to tackle the semantic inconsistency problem between the augmented views when pre-training a discriminative user model based on user behavior sequences.
> 2.  **The definitions of the target ranking order are completely different.** Yu et al [49] aim to model the order of the user's preference difference between (I) an observed item and an unobserved item, (II) two unobserved items, and (III) two observed items, while we train the user model to capture the similarity orders between (I) the implicitly augmented views, (II) the explicitly augmented views, and (III) views from other users.
> 3.  **How to construct the training triplet is different.** Yu et al [49] just randomly sample items from the corresponding item set, while we design an explicit hard negative sampling strategy to facilitate model training, which selects the pair with the smallest similarity difference for each pairwise ranking order.
> 4.  **How to fuse the two learned pairwise ranking order is different.** Yu et al [49] use a fixed and unified hyperparameter $\lambda$ in the original multiple pairwise ranking loss, while our augmentation-adaptive fusion mechanism takes the distinct impacts of data augmentation on different behavior sequences into account and employs a dynamic coefficient $\lambda_i$ for each training sample $S_i$. The value of $\lambda_i$ is calculated based on the estimated semantic similarity between the augmented views along the training procedure.
>
> The only connection between our work and Yu et al [49] is that we both try to simultaneously learn a ranking order between three terms. That's why we adopt the Multiple Pairwise Ranking loss for model training in our work. We will highlight the difference between our work and Yu et al [49] in the revised version of our paper.
>
> Second, the main contributions of our paper are as follows:
>
> 1. We identify the semantic inconsistency problem when applying contrastive learning to user behavior sequences.
> 2. To tackle this problem, we escape the existing contrastive learning framework and propose a new augmentation-adaptive self-supervised ranking task. Instead of simply training the model to distinguish the positive augmented views from the negative ones as contrastive learning, we train the user model to capture a more precise and realistic similarity order between the implicitly augmented view, the explicitly augmented view, and views from other users.
> 3. Different from contrastive learning which simply maximizes the similarity between the augmented views for every sample in a fixed way, we further design an augmentation-adaptive fusion mechanism that adaptively adjusts the similarity order constraint applied to each sample based on the semantic similarity between the augmented views.
> 4. Extensive experiments on both public and industrial datasets verify that our AdaptSSR can be applied to different kinds of user behavior sequences and bring significant performance improvement to various downstream tasks.
>
> To our best knowledge, our work is the first to tackle the semantic inconsistency problem when applying contrastive learning in the user modeling domain, and our method can be further generalized to other domains where augmentation choices are not straightforward or could alter the semantics of the data. We will highlight our contributions in the revised version of our paper.
>
> **Limitation 1**: The authors have not discussed the limitations and broader societal impacts in the paper.
>
> **Response**: Thanks for pointing it out. We have discussed the limitations of our work in the Appendix due to the space limit. We will put them in the main content in the revised version of our paper.

---

> > ### Comment · Reviewer_toy8 · 2023-08-20
> >
> > The authors have addressed my questions. I'm maintaining my previous score.

---

> > > ### Author Response · Authors · 2023-08-21
> > >
> > > Thank you for your valuable feedback. We are delighted to know that our responses have addressed your concerns. We sincerely appreciate your time and efforts in helping us improve the quality of our work. Thank you very much.

---

> ### Author Response · Authors · 2023-08-17
>
> Dear Reviewer toy8,
>
> Thank you for your insightful review comments. Considering the deadline for the author-reviewer discussion period is approaching, we are writing to follow up on our previous rebuttal submission and inquire if there are any remaining concerns or questions that we can address to improve the quality of our paper. We are open to constructive feedback and eager to work with you to improve our work.
>
> Thanks,
>
> Authors.

---

### Official Review · Reviewer_8cxi · 2023-07-03

**Soundness:** 1 poor
**Presentation:** 1 poor
**Contribution:** 2 fair
**Rating:** 4
**Confidence:** 5

**Summary:**

Recent studies have explored pre-training user models with contrastive learning tasks to address data sparsity issues in user-oriented tasks. However, existing augmentation methods may introduce noisy or irrelevant interests, leading to negative transfer. To overcome this, a new approach called Augmentation-Adaptive Self-Supervised Ranking (AdaptSSR) is proposed, which replaces contrastive learning with a multiple pairwise ranking loss. An augmentation-adaptive fusion mechanism is also introduced to combine learned ranking orders based on the similarity between augmented views. Extensive experiments demonstrate the effectiveness of AdaptSSR across various tasks and datasets.

**Strengths:**

1. The paper's motivation is reasonable, directly adopting contrastive learning may lead to consistency problems in recommendations.
2. This paper proposes a novel approach to combine implicit and explicit augmentations.

**Weaknesses:**

1. The main contribution of this paper is adding an order constraint in the loss function,
   which is a rather incremental modification of existing contrastive learning framework.
   The main idea of paper is a fusion of explicit augmentation and implicit augmentation by the loss function.
   Thus, the novelty is limited.

2. It's unclear whether the added constraint is necessary. Since "$u$ and $u^+$ originate from exactly the same input behavior sequence" as the authors commented in line 123, I think $sim(u, u^+)\ge sim(u, u^-)$ and $sim(u, u^+) \ge sim(u, \hat{u}$ should always hold. I don't understand why we need the $sim(u, u^+)$ term here. Without the $sim(u, u^+)$ term, the proposed method reduces to common contrastive learning.

3. Even if the constraint is meaningful, the authors' analysis cannot convince me why such constraint may help generalization. Why may $sim(u, \hat{u}) > sim(u, u^+)$ harm the downstream performance? Intuitively, suppose the objective of original contrastive learning is overly strong, we should loose the constraints. For example, $sim(u, \hat{u}) \ge sim(u, u^-) - \epsilon$. However, the authors make the constraints even stronger by adding another constraint term. This does not make sense to me.

---

Edit after rebuttal: The authors' response resolved my primary concern about technical correctness. I'd like to raise my score to a borderline reject regarding the novelty of this work.


**Questions:**

1. Please explain what is implicit augmentation, how is $u^+$ generated, and why equation (1) should hold.
2. The authors introduce a Augmentation-Adaptive Fusion coefficient $\lambda$ needs further discussion. Is the $\lambda$ fixed during training, i.e. stop_gradient($\lambda$), so that the gradient calculation does not involve $\lambda$?

**Limitations:**

Please address the issues highlighted in the Weaknesses section.

---

> ### Author Rebuttal · Authors · 2023-08-09
>
> We appreciate your time in reviewing our paper. However, we suspect that there may be certain misconceptions. In order to address your concerns, we provide detailed, point-by-point responses as follows. **Due to the character limit of each rebuttal, more responses are provided in the global response.**
>
> **Weakness 1**: The main contribution of this paper is adding an order constraint in the loss function, which is a rather incremental modification of the existing contrastive learning framework. The main idea of the paper is a fusion of explicit augmentation and implicit augmentation by the loss function. Thus, the novelty is limited.
>
> **Response**: We appreciate your feedback. Indeed, as you have pointed out, the introduction of an order constraint represents one facet of our contributions. However, it is important to clarify that this is not the sole contribution of our work and our method differs from contrastive learning in several aspects. Specifically, the main contributions of our paper are as follows:
>
> 1. We identify the semantic inconsistency problem when applying contrastive learning to user behavior sequences.
> 2. To tackle this problem, we escape the existing contrastive learning framework and propose a new augmentation-adaptive self-supervised ranking task. Instead of simply training the model to distinguish the positive augmented views from the negative ones as contrastive learning, we train the user model to capture a more precise and realistic similarity order between the implicitly augmented view, the explicitly augmented view, and views from other users.
> 3. Different from contrastive learning which simply maximizes the similarity between the augmented views for every sample in a fixed way, we further design an augmentation-adaptive fusion mechanism that adaptively adjusts the similarity order constraint applied to each sample based on the semantic similarity between the augmented views.
> 4. Extensive experiments on both public and industrial datasets verify that our AdaptSSR can be applied to different kinds of user behavior sequences and bring significant performance improvement to various downstream tasks.
>
> Therefore, our method differs from contrastive learning in terms of the constraint applied to each sample and the design of the loss function. To our best knowledge, our work is the first to tackle the semantic inconsistency problem when applying contrastive learning in the user modeling domain, and our method can be further generalized to other domains where augmentation choices are not straightforward or could alter the semantics of the data. Overall, we think the contribution and novelty of our work are non-trivial.
>
> **Weakness 2**: It's unclear whether the added constraint is necessary. Since "$u$ and $u^+$ originate from exactly the same input behavior sequence" as the authors commented in line 123, I think $sim(u,u^+)\geq sim(u,u^-)$ and $sim(u,u^+)\geq sim(u,\hat{u})$ should always hold. I don't understand why we need the $sim(u,u^+)$ term here. Without the $sim(u,u^+)$ term, the proposed method reduces to common contrastive learning.
>
> **Response**: First, it is essential to clarify that although $u$ and $u^+$ originate from exactly the same input behavior sequence, different independently sampled dropout masks are applied in the user model, which adds distinct noise to the input sequence in the feature space. Therefore, $u$ and $u^+$ are different, and the pairwise similarity order $sim(u,u^+)\geq sim(u,u^-)$ and $sim(u,u^+)\geq sim(u,\hat{u})$ do not always hold.
>
> In addition, the $sim(u,u^+)$ term is necessary as it works as the upper bound of $sim(u,\hat{u})$. Without this term, our method will apply the same constraint as contrastive learning and the model will directly maximize $sim(u,\hat{u})$ for every sample while neglecting the potential semantic inconsistency between the augmented views. Besides, our augmentation-adaptive fusion mechanism can further adjust $sim(u,\hat{u})$ between the upper bound $sim(u,u^+)$ and the lower bound $sim(u,u^-)$ properly for each sample based on the semantic similarity between the augmented views. Moreover, from the results in Table 2, we can find that our AdaptSSR consistently outperforms existing contrastive learning-based pre-training methods (e.g., CCL, CL4SRec, CoSeRec), which further verifies the effectiveness of the additional $sim(u,u^+)$ term.
>
> **Weakness 3.1**: Even if the constraint is meaningful, the authors' analysis cannot convince me why such constraint may help generalization. Why may $sim(u,\hat{u})>sim(u,u^+)$ harm the downstream performance?
>
> **Response**: We acknowledge that we may not have made this sufficiently clear in our paper, and we appreciate the opportunity to clarify. The reason why we require $sim(u,u^+)\geq sim(u,\hat{u})$ is that the difference between $u$ and $u^+$ is only caused by the different dropout masks applied by the implicit augmentation, while the difference between $u$ and $\hat{u}$ is caused by both the implicit augmentation and the explicit augmentation which directly modifies the input behavior sequence on the data level. If $sim(u,\hat{u})>sim(u,u^+)$, it means the user model cannot well capture the characteristics and interests of the user from the behavior sequence, and the generated user embedding cannot correctly reflects the similarity between different users. Thus, it will transfer incorrect prior knowledge to the downstream task and degrade the model performance, especially when the downstream task faces a data sparsity problem.

---

> ### Author Response · Authors · 2023-08-14
>
> We appreciate your acknowledgment of our response and we are delighted to know that our replies have clarified the technical correctness of our paper. To further address your concern regarding the novelty of our work, we'd like to highlight that our work is the first to identify and tackle the semantic inconsistency problem when applying contrastive learning in the user modeling domain. Our AdaptSSR escapes the existing contrastive learning framework and provides a new pre-training schema that can be further generalized to other domains where augmentation choices are not straightforward or could alter the semantics of the data. We hope the contribution of our work can be recognized. If you have any other concerns, please feel free to reach out to us. We assure you that we will do our best to resolve any concerns you may have about this paper.

---

### Official Review · Reviewer_P6ts · 2023-07-04

**Soundness:** 3 good
**Presentation:** 3 good
**Contribution:** 3 good
**Rating:** 6
**Confidence:** 4

**Summary:**

This paper proposes Augmentation-Adaptive Self-Supervised Ranking (AdaptSSR), a new user model pre-training paradigm, which alleviates the requirement of semantic consistency between the augmented views while pre-training a discriminative user model. Conventional methods assume that different views of the same behaviour sequence constructed via data augmentation are semantically consistent, while in practice existing augmentation methods tend to lose certain interests of the user or introduce noisy interests that the user does not have. AdaptSSR addresses this issue by adopting a multiple pairwise ranking loss which trains the user model to capture the similarity orders between the explicitly augmented views, the implicitly augmented views, and views from other users. An explicit hard negative sampling strategy and an augmentation-adaptive fusion mechanism are also introduced to facilitate model training. Extensive experiments on both public and industrial datasets verify the effectiveness of AdaptSSR.

**Strengths:**

- The proposed approach is technically sound and the empirical results validates the effectiveness of the method.
- The paper is well-written with very clear figures.
- Code is available which makes it easy to reproduce the results.

**Weaknesses:**

An important hyperparameter sensitivity analysis is missing: how does the value of $\lambda$ affect the model performance? Compared with existing models, AdaptSSR introduces an extra SimCSE-inspired implicit augmentation approach. It remains unclear in the paper if the performance improvement is primarily due to the introduction of implicit data augmentation.

**Questions:**

Only three random augmentation operators are used in the paper. Does introducing more augmentation operators help improve the performance? Do informative augmentation operations introduced by CoSeRec help improve AdaptSSR's performance?

**Limitations:**

The extra computational cost introduced by AdaptSSR is not analyzed in the paper, it would be useful if the authors can demonstrate the tradeoff between training time and performance for various models.

---

> ### Author Rebuttal · Authors · 2023-08-09
>
> We very much appreciate your positive opinions on the contribution and presentation of this paper. We also thank you for the valuable comments and our detailed responses are as follows.
>
> **Weakness 1**: An important hyperparameter sensitivity analysis is missing: how does the value of $\lambda$ affect the model performance?
>
> **Response**: Many thanks for this good question. $\lambda$ is a critical hyperparameter in our method since it controls how the two learned pairwise ranking orders: $sim(u,u^+)\geq sim(u,\hat{u})$ and $sim(u,\hat{u})\geq sim(u,u^-)$ are fused. As we mentioned in Line 171 of our paper, the effects of data augmentation vary significantly across diverse behavior sequences. As a result, a fixed and unified $\lambda$ is not enough to combine the two pairwise ranking orders properly for different samples. That's why we design an augmentation-adaptive fusion mechanism that replaces the hyperparameter $\lambda$ with a dynamic coefficient $\lambda_i$ for each training sample $S_i$. The value of $\lambda_i$ is calculated based on the average similarity between the user representations generated from the augmented views $\hat{S}_i$ and $\tilde{S}_i$ (Equation (7)) along the training procedure. As a result, we no longer need to manually set the value of $\lambda$, and the effectiveness of our augmentation-adaptive fusion mechanism has been verified in Section 4.4. We will highlight how we replace $\lambda$ with $\lambda_i$ in the revised version of our paper.
>
> **Weakness 2**: Compared with existing models, AdaptSSR introduces an extra SimCSE-inspired implicit augmentation approach. It remains unclear in the paper if the performance improvement is primarily due to the introduction of implicit data augmentation.
>
> **Response**: Many thanks for pointing it out. Actually, we have taken the impact of introducing extra implicit augmentation into account and compared our AdaptSSR with CLUE [1] in our experiments. It is worth mentioning that CLUE shares the same model structure with AdaptSSR but only uses implicitly augmented views for contrastive pre-training. From the results in Table 2, we can find that our AdaptSSR consistently outperforms CLUE by a large margin on various downstream tasks. We argue that it is because the implicit augmentation caused by the dropout mask alone is too weak. The user model can easily distinguish the positive samples from others, thus providing limited knowledge for downstream tasks. Such results illustrate that introducing extra implicit augmentation is not the primary reason for performance improvement. We will highlight the comparison between AdaptSSR and CLUE in the revised version of our paper.
>
> [1] Mingyue Cheng et al. Learning Transferable User Representations with Sequential Behaviors via Contrastive Pre-training. ICDM 2021.
>
> **Question 1**: Only three random augmentation operators are used in the paper. Does introducing more augmentation operators help improve the performance? Do informative augmentation operations introduced by CoSeRec help improve AdaptSSR's performance?
>
> **Response**: Thanks for this insightful question. As we mentioned in Line 147 of our paper, our method can be combined with various data augmentation methods. We have evaluated the performance of AdaptSSR when combining it with several existing data augmentation methods in Section 4.3, including these informative augmentation operators introduced by CoSeRec. From the results in Figure 3 (the three dashed lines on top), we can find that our method achieves similar performance when combined with different data augmentation methods. We argue that this is because our augmentation-adaptive fusion mechanism can always properly combine the learned pairwise ranking orders based on the estimated similarity between the explicitly augmented views constructed by different augmentation methods, which leads to similar model performance. We will add this analysis to the revised version of our paper. Once again, we express our appreciation for your insightful feedback.
>
> **Limitations 1**: The extra computational cost introduced by AdaptSSR is not analyzed in the paper, it would be useful if the authors can demonstrate the tradeoff between training time and performance for various models.
>
> **Response**: We extend our gratitude for your insightful suggestions. The average pre-training time of different methods on the TTL dataset and the App dataset are listed in the following table. Our observations indicate that the average pre-training time of our AdaptSSR is similar to that of existing contrastive learning-based methods, such as CL4SRec and CoSeRec. Although AdaptSSR requires inputting each behavior sequence into the model twice, we implement it by duplicating the input sequences in the batch size dimension. Thus, we only need to input all the sequences into the model once, which can be well parallelized by the GPU. Besides, for each pairwise ranking order, our hard negative sampling strategy only selects the pair with the smallest similarity difference to compute the multiple pairwise ranking loss, which avoids the costly softmax operation and cross-entropy loss calculation in the existing contrastive learning task. As a result, our AdaptSSR will not greatly increase the overall computational cost while bringing performance improvement to various downstream tasks. We will add the result and analysis to the revised version of our paper.
> \begin{array}{ccc}\hline\text{Pre-train Method}&\text{TTL}&\text{App}\\\\\hline\text{PeterRec}&2.927±0.022\text{h}&1.537±0.010\text{h}\\\\
> \text{PTUM}&2.015±0.009\text{h}&2.055±0.018\text{h}\\\\\text{CLUE}&1.453±0.016\text{h}&1.633±0.015\text{h}\\\\
> \text{IDICL}&1.257±0.021\text{h}&1.162±0.020\text{h}\\\\\text{CL4SRec}&1.868±0.013\text{h}&2.081±0.017\text{h}\\\\
> \text{CoSeRec}&1.902±0.015\text{h}&2.104±0.023\text{h}\\\\\text{DuoRec}&1.535±0.020\text{h}&1.658±0.015\text{h}\\\\
> \text{AdaptSSR}&1.539±0.017\text{h}&1.830±0.012\text{h}\\\\\hline\end{array}

---

> > ### Comment · Reviewer_P6ts · 2023-08-17
> >
> > After reading the authors' rebuttal, most of my concerns have been properly addressed. Therefore I have raised my score. Thanks for the detailed explanation.

---

> > > ### Author Response · Authors · 2023-08-17
> > >
> > > Thank you for your appreciation and valuable feedback. We will incorporate the additional results and analysis into the final version of our paper. We sincerely appreciate your time and efforts in helping us improve the quality of our work.

---

### Official Review · Reviewer_oeWK · 2023-07-06

**Soundness:** 3 good
**Presentation:** 3 good
**Contribution:** 3 good
**Rating:** 6
**Confidence:** 4

**Summary:**

The authors tackle the problem of doing self-supervised learning for user modeling.  Inspired by the successes of contrastive learning approaches in the image setting, they adapt contrastive learning to the user modeling setting.  However, in user modeling the augmentations typically used are not very suitable for contrastive learning because they can change the semantics of the data, thus forcing similarity between augmented views can be problematic.  They instead produce three views: the anchor, a similar "implicitly" augmented view, and a less similar "explicitly" augmented view.  The implicitly augmented view is trained to be more similar to the anchor than the explicitly augmented view.  This escapes the problematic similarity training that plain contrastive learning would have in user modeling.

**Strengths:**

1.  The paper was well-written and the diagrams were easy to understand.  It made the paper easy to read and review.  The contributions were clearly stated and explained in the paper.

2.  The method is novel and original as far as I know.  This method could be generalizable to other domains where augmentation choices are not straightforward and could alter the semantics of the data.

3.  The method is well-designed: the ranking loss does help mitigate the "make semantically different augmented views the same" problem, and furthermore helps balance the focus of the loss between the implicit vs explicit contrast and the explicit vs other user contrast.

4.  The improvements in the empirical results are consistent and seem to be significant.


**Weaknesses:**

1.  Going back to the example where the user behavior is represented by a sequence of images, is it possible to just do per-image augmentation (choices for these exist and are widely used) and then perform a typical InfoNCE style contrastive loss on the user embeddings?  For text one could do something similar using masking augmentations and such.  I did not see a comparison to this baseline and I wonder how well it would perform.  I think this is something that would be critical to compare against.

2.  While the paper is written well, I think the paper should define what user modeling is and what the downstream tasks are earlier in the paper (or in the abstract).  For a while it was not clear to me what problem the paper was trying to solve, as someone who has not worked on user modeling.

3.  Adding error bars into the results tables would help in understanding the significance of the results.

**Questions:**

Questions are listed in the weaknesses.

**Limitations:**

Seems sufficient.

---

> ### Author Rebuttal · Authors · 2023-08-09
>
> We really appreciate your careful reading and constructive comments. We also very much appreciate your acknowledgment that our proposed method is novel and well-designed. Following are our detailed responses to your comments.
>
> **Weakness 1**: Going back to the example where the user behavior is represented by a sequence of images, is it possible to just do per-image augmentation and then perform a typical InfoNCE style contrastive loss on the user embeddings? For text, one could do something similar using masking augmentations and such. I did not see a comparison to this baseline and I wonder how well it would perform. I think this is something that would be critical to compare against.
>
> **Response**: Thanks for your insightful comment. Integrating multimodal information related to user behaviors into user model pre-training is an interesting and promising direction. However, it is essential to clarify that the images depicted in Figure 1 are just used as illustrations of what each news article clicked by the user is about, so the readers can understand the impact of different augmentation methods on the behavior sequence more intuitively.
>
> In this work, our methodology aligns with the prevalent settings in many previous works [1-4] and real-world scenarios, where each behavior is simply represented by an ID. We refrain from using any ancillary information related to user behaviors (such as the image or text description), so we cannot perform per-image or per-text augmentation. Sorry for the unclear description. We will provide a more explicit explanation of the images presented in Figure 1 in the revised version of our paper.
>
> [1] Fajie Yuan et al. Parameter-Efficient Transfer from Sequential Behaviors for User Modeling and Recommendation. SIGIR 2020. \
> [2] Mingyue Cheng et al. Learning Transferable User Representations with Sequential Behaviors via Contrastive Pre-training. ICDM 2021. \
> [3] Xu Xie et al. Contrastive Learning for Sequential Recommendation. ICDE 2022. \
> [4] Shuqing Bian et al. Contrastive Curriculum Learning for Sequential User Behavior Modeling via Data Augmentation. CIKM 2021.
>
> **Weakness 2**: While the paper is written well, I think the paper should define what user modeling is and what the downstream tasks are earlier in the paper (or in the abstract). For a while, it was not clear to me what problem the paper was trying to solve, as someone who has not worked on user modeling.
>
> **Response**: We appreciate your constructive suggestion. Indeed, user modeling aims to capture the user's characteristics or interests for a specific user-oriented task (e.g., personalized recommendation and click-through rate prediction) and encode them into a dense representation with a user representation model. As existing supervised user modeling methods tend to suffer from the data sparsity problem, our work aims to pre-train the user representation model on massive unlabeled user behavior data, which enables it to extract users' general characteristics or interests from their historical behaviors and can be transferred to benefit various downstream tasks. We will reorganize our paper and clarify the definition in the Abstract and Introduction section. Thank you again for your constructive comment.
>
> **Weakness 3**: Adding error bars into the results tables would help in understanding the significance of the results.
>
> **Response**: Thanks for the advice. We have added the standard deviation of each experiment to Table 2 in our paper, which is shown as follows (highlight in bold). Our further t-test results show that compared with the second-best method, the improvements of our AdaptSSR are significant at $p<0.01$ on every downstream task. We will add the results to the revised version of our paper.
> \begin{array}{c|cc|cc|cc|cc|cc|cc}\hline\text{Pre-train}&\mathcal{T}_1&&\mathcal{T}_2&&\mathcal{T}_3&&\mathcal{T}_4&&\mathcal{T}_5&&\mathcal{T}_6&&\\\\
> \text{Method}&\text{Acc}&\text{Impr\\%}&\text{Acc}&\text{Impr\\%}&\text{NDCG\@10}&\text{Impr\\%}&\text{NDCG\@10}&\text{Impr\\%}&\text{AUC}&\text{Impr\\%}&\text{AUC}&\text{Impr\\%}&\\\\\hline\text{None}&0.6287\bf{\pm0.0005}&-&0.5224\bf{\pm0.0016}&-&0.0199\bf{\pm0.0003}&-&0.0287\bf{\pm0.0007}&-&0.7863\bf{\pm0.0006}&-&0.7514\bf{\pm0.0014}&-&\\\\\text{PeterRec}&0.6362\bf{\pm0.0011}&1.19&0.5314\bf{\pm0.0007}&1.72&0.0237\bf{\pm0.0002}&19.12&0.0306\bf{\pm0.0008}&6.37&0.7961\bf{\pm0.0013}&1.25&0.7604\bf{\pm0.0010}&1.20&\\\\\text{PTUM}&0.6321\bf{\pm0.0014}&0.54&0.5305\bf{\pm0.0004}&1.55&0.0229\bf{\pm0.0003}&14.65&0.0296\bf{\pm0.0003}&3.17&0.7948\bf{\pm0.0011}&1.08&0.7582\bf{\pm0.0013}&0.90&\\\\\text{CLUE}&0.6338\bf{\pm0.0010}&0.81&0.5323\bf{\pm0.0005}&1.90&0.0238\bf{\pm0.0002}&19.62&0.0305\bf{\pm0.0021}&6.02&0.7990\bf{\pm0.0006}&1.62&0.7603\bf{\pm0.0016}&1.18&\\\\\text{CCL}&0.6376\bf{\pm0.0011}&1.42&0.5337\bf{\pm0.0009}&2.16&0.0243\bf{\pm0.0002}&21.93&0.0332\bf{\pm0.0013}&15.74&0.8022\bf{\pm0.0007}&2.02&0.7735\bf{\pm0.0010}&2.94&\\\\\text{IDICL}&0.6388\bf{\pm0.0004}&1.61&0.5345\bf{\pm0.0005}&2.32&0.0246\bf{\pm0.0002}&23.63&0.0342\bf{\pm0.0004}&19.01&0.8034\bf{\pm0.0005}&2.17&0.7792\bf{\pm0.0008}&3.70&\\\\\text{CL4SRec}&0.6371\bf{\pm0.0014}&1.34&0.5343\bf{\pm0.0005}&2.28&0.0241\bf{\pm0.0003}&21.12&0.0329\bf{\pm0.0006}&14.42&0.8014\bf{\pm0.0008}&1.92&0.7702\bf{\pm0.0005}&2.50&\\\\\text{CoSeRec}&0.6389\bf{\pm0.0003}&1.62&0.5353\bf{\pm0.0009}&2.47&0.0244\bf{\pm0.0002}&22.53&0.0333\bf{\pm0.0005}&15.77&0.8048\bf{\pm0.0006}&2.35&0.7771\bf{\pm0.0009}&3.42&\\\\\text{DuoRec}&0.6350\bf{\pm0.0009}&1.00&0.5326\bf{\pm0.0006}&1.95&0.0239\bf{\pm0.0001}&20.12&0.0311\bf{\pm0.0016}&8.32&0.8003\bf{\pm0.0009}&1.78&0.7685\bf{\pm0.0009}&2.28&\\\\\text{AdaptSSR}&0.6553\bf{\pm0.0004}&4.23&0.5441\bf{\pm0.0002}&4.15&0.0261\bf{\pm0.0003}&30.71&0.0373\bf{\pm0.0003}&29.77&0.8230\bf{\pm0.0003}&4.67&0.7992\bf{\pm0.0005}&6.36&\\\\\hline\end{array}

---

> > ### Comment · Reviewer_oeWK · 2023-08-15
> > **Thanks for the rebuttal**
> >
> > I will keep my score after reading the rebuttals and the other reviews.  Thanks for adding significance test results.

---

> > > ### Author Response · Authors · 2023-08-16
> > >
> > > We are sincerely grateful for your appreciation and valuable comments. We will revise our paper accordingly to incorporate your suggestions and the additional results. We really appreciate your time and efforts in helping us improve the quality of our work. Thank you very much.

---

### Author Rebuttal · Authors · 2023-08-09

We sincerely thank all the reviewers for their appreciation and constructive comments. We have provided detailed responses to each reviewer's concerns and questions in the following rebuttals. We hope our responses will address your concerns and strengthen our paper. We are happy to respond to any new questions during the discussion period.

---

**More responses to Reviewer 8cxi**

**Weakness 3.2**: Intuitively, suppose the objective of original contrastive learning is overly strong, we should loose the constraints. For example, $sim(u,\hat{u})\geq sim(u,u^-)-\epsilon$. However, the authors make the constraints even stronger by adding another constraint term. This does not make sense to me.

**Response**: We appreciate your insightful thinking. However, the primary issue of contrastive learning does not lie in an overly strong objective. The problem is that it imposes a fixed and inaccurate constraint $sim(u,\hat{u})\geq sim(u,u^-)$ on every sample while the semantic similarity between the augmented views cannot be guaranteed. The InfoNCE-style contrastive loss will simply maximize $sim(u,\hat{u})$ no matter whether the augmented views are similar or not, which will lead to a negative transfer for the downstream task. Our method applies an adaptive similarity order constraint to each sample by adjusting $sim(u,\hat{u})$ between the upper bound $sim(u,u^+)$ and the lower bound $sim(u,u^-)$ based on the semantic similarity between the augmented views. We will refine our writing to make our motivation clearer.

**Question 1**: Please explain what is implicit augmentation, how is $u^+$ generated, and why equation (1) should hold.

**Response**: We appreciate your comment. We will respond to each of your questions in a sequential manner.

- As we mentioned in Line 99 of our paper, the implicit augmentation is performed via the dropout module in the model, which adds noise to the input sequence in the feature space.
- As we mentioned in Line 148 of our paper, given a behavior sequence $S$, we input it into the model twice with different independently sampled dropout masks and denote the generated user representations as $u$ and $u^+$.
- Since the difference between $u$ and $u^+$ is only caused by the different dropout masks applied by the implicit augmentation, while the difference between $u$ and $\hat{u}$ is caused by both the implicit augmentation and the explicit augmentation which directly modifies the input behavior sequence on the data level, we require the model to capture the similarity order $sim(u,u^+)\geq sim(u,\hat{u})$. Similarly, the difference between $u$ and $u^-$ is caused by both the implicit augmentation and the distinct interests of different users. Therefore, we require the model to capture the similarity order $sim(u,u^+)\geq sim(u,u^-)$. Since the explicit augmentation modifies $S$ at the data level and the semantic consistency between $S$ and $\hat{S}$ cannot be guaranteed, $sim(u,\hat{u})$ should be placed between $sim(u,u^+)$ and $sim(u,u^-)$, which leads to the final similarity order $\Gamma: sim(u,u^+)\geq sim(u,\hat{u})\geq sim(u,u^-)$ (Equation (1)). Our goal is pre-training the user model to adjust $sim(u,\hat{u})$ properly between $sim(u,u^+)$ and $sim(u,u^-)$ based on the semantic similarity between the augmented views.

**Question 2**: The authors introduce an Augmentation-Adaptive Fusion coefficient $\lambda$ that needs further discussion. Is the $\lambda$ fixed during training, i.e.,$\operatorname{stop\\_gradient}(\lambda)$, so that the gradient calculation does not involve $\lambda$?

**Response**: As we mentioned in Line 175 of our paper, our augmentation-adaptive fusion mechanism replaces the fixed and unified hyperparameter $  \lambda $ with a dynamic coefficient $\lambda_i$ for each training sample $S_i$. The value of $\lambda_i$ is dynamically calculated based on the average similarity between the user representations generated from the augmented views $\hat{S}_i$ and $\tilde{S}_i$ (Equation (7)) along the training procedure. It is not a learnable parameter so it is not involved in the gradient calculation. We will highlight how $\lambda_i$ is dynamically calculated in the revised version of our paper.

---

### Decision · Program_Chairs · 2023-09-21

**Decision:**

Accept (poster)

**Comment:**

The paper proposes and analyzes AdaptSSR, a self-supervised learning (SSL) technique for user modeling from user behavior sequences to address issues like data sparsity in real-world user platforms. Experimental results demonstrate AdaptSSR's good effectiveness across tasks and datasets, from both public domain and industry. However, some reviewers raised concerns regarding the method's clarity. For example, the notations used are not very intuitive and it is hard to understand how different implicitly augments views are generated. I highly encourage the authors to update their manuscript using reviewer suggestions.